# Test-retest reliability and minimal detectable change of Dutch-Flemish Patient Reported Outcomes Measurement Information System (PROMIS®) Computerized Adaptive Tests for musculoskeletal disorders

R. M. Arensman[1,2]*, E. J. A. Haan[1,3,4,5], C. B. Terwee[4,5], J. van Rosmalen[6], H. Wittink[1], H. Kiers[1,7]

1 Research group Clinical Decision Making in Movement Care, HU University of Applied Sciences Utrecht, Utrecht, The Netherlands, 2 Institute for Human Movement Studies, HU University of Applied Sciences Utrecht, Utrecht, The Netherlands, 3 Center for Physical Therapy Research and Innovation in Primary Care, Julius Health Care Centers, Utrecht, The Netherlands, 4 Department of Epidemiology and Data Science, Amsterdam UMC, Vrije Universiteit, Amsterdam, The Netherlands, 5 Amsterdam Public Health Research Institute, Methodology, Amsterdam, The Netherlands, 6 Julius Center for Health Sciences and Primary Care, University Medical Center Utrecht, Utrecht University, Utrecht, The Netherlands, 7 Department of Human Movement Sciences, Vrije Universiteit Amsterdam, Amsterdam, the Netherlands

* remco.arensman@hu.nl

## Abstract

### Objective

To examine the test-retest reliability and minimal detectable change (MDC) of the Dutch-Flemish Patient Reported Outcomes Measurement Information System (DF-PROMIS) Pain Interference (PI) v1.1, Physical Function (PF) v1.2, and Upper Extremity (UE) v2.0 computerized adaptive tests (CATs) in patients with musculoskeletal conditions receiving physical therapy in primary care.

### Design

Observational cohort study.

### Methods

Patients with musculoskeletal conditions of the spine or upper extremity were recruited from fourteen physical therapy practices. Participants completed DF-PROMIS CATs at baseline and again three to fourteen days later. Test-retest reliability was evaluated using the intraclass correlation coefficient (ICC) (two-way random effects, absolute agreement) and minimal detectable change (MDC). Reliability at the participant-level was visually represented by plotting test-retest scores with corresponding 95% confidence intervals (CIs).

**Data availability statement:** All files (dataset, scripts, metadata) used for the analysis are available from https://doi.org/10.34894/6ABGM0.

**Funding:** This study is co-funded by the Taskforce for Applied Research SIA (RAAK.MKB13.025), part of the Dutch Research Council (NWO).

**Competing interests:** I have read the journal's policy and the authors of this manuscript have the following competing interests: author C.B. Terwee, PhD, is past board member of the PROMIS Health Organization and representative of the Dutch-Flemish PROMIS National Center.

## Results

Data from 225 patients were analyzed. The DF-PROMIS CATs demonstrated sufficient test-retest reliability, with ICC values ranging from 0.79 to 0.91. MDC values ranged from 4.80 to 6.08 across all measurements. Participant-level reliability was high (0.9–0.95) for most measurements but lower for scores further from the mean. The 95% CIs for test-retest measurements overlapped in 95.3% of measurement pairs.

## Conclusion

The DF-PROMIS PF, UE, and PI domain CATs demonstrated sufficient reliability and precision in patients with musculoskeletal conditions receiving physical therapy in primary care practices. Future research should focus on implementing DF-PROMIS CATs in clinical practice, examining their responsiveness, and evaluating their feasibility. Adoption of DF-PROMIS domains as outcomes in intervention studies and clinical practice will enhance interpretability and comparability of results across different patient groups.

## Introduction

Musculoskeletal conditions are the leading contributor to disability worldwide [1,2] and are the highest contributor to the global need for rehabilitation [3]. Rehabilitation can potentially reduce the enormous financial costs related to disability due to musculoskeletal conditions [4] and is often provided by physical therapists. In the Netherlands, physical therapy is provided mainly in primary care physical therapy practices by approximately 21.600 therapists [5]. The most common conditions treated by physical therapists in primary care physical therapy practices in The Netherlands are musculoskeletal disorders of the spine and upper extremity [6].

The main purpose of physical therapy treatment for patients with musculoskeletal conditions is to provide patient-centered care within a biopsychosocial model tailored to the individual's personal goals and preferences [7]. To this end, physical therapy treatments aim to improve physical function or to reduce the interference of pain with daily functioning in patients. Patient Reported Outcome Measures (PROMs) help to operationalize these principles by systematically capturing the patient's own perspective on their symptoms, functioning, and quality of life, thereby ensuring that biological, psychological, and social dimensions of health are included in clinical decision-making [8]. In this way, PROMs support physical therapists in their clinical reasoning related to diagnosis, treatment, and evaluation [8]. A recent systematic review found evidence that feedback from PROMs can improve quality of life, patient-provider communication, and disease control [9]. However, physical therapists find it difficult to select appropriate PROMs for each patient, due to the large number of PROMs available [10], and lack of knowledge regarding the assessment of the quality of available instruments [11]. Despite attempts to support PROM selection by

clinicians [12,13], physical therapists indicated the need for a core set of PROMs with clear instructions regarding their application, scoring, and interpretation [11].

In the Netherlands, the Dutch-Flemish translation of the Patient Reported Outcomes Measurement Information System (PROMIS) developed by a US consortium of research groups, funded by the National Institutes of Health (NIH) [14,15] was recommended to standardize measurement of PROMs across different patient groups [16]. PROMIS instruments were selected because they provide precise and efficient assessment of relevant health outcomes, with standardized scores that enable comparability across studies and patient populations [14,17]. Furthermore, PROMIS is a valid and reliable measurement system, which was developed using Item Response Theory (IRT) [18]. PROMIS consists of a collection of IRT-based item banks for relevant domains in healthcare, such as physical function (PF), pain interference (PI), and fatigue [15]. From these item banks, Computerized Adaptive Tests (CATs) [19] were developed, which select the most informative questions for each individual based on their responses. This results in patient-specific assessments that require fewer items to complete, while achieving equal or greater measurement precision compared to traditional fixed item questionnaires [14,20]. PROMIS scores are reported as T-scores, standardized to a reference population with a mean of 50 and a standard deviation of 10 [18]. This allows patients' outcomes to be interpreted relative to the general population and facilitates comparisons across different patient groups using a common metric.

Several PROMIS item banks relevant to use for the evaluation of treatment effects in patients with musculoskeletal conditions receiving physical therapy have been validated in clinical samples [15]. These item banks are the Dutch-Flemish PROMIS Pain Interference (DF-PROMIS-PI) v1.1 item bank (patients with chronic pain) [21], the Dutch-Flemish PROMIS Physical Function (DF-PROMIS-PF) v1.2 item bank (patients with chronic pain or receiving physical therapy) [22,23], and the Dutch-Flemish PROMIS Upper Extremity (DF-PROMIS-UE) v2.0 item bank (patients with UE disorders) [24–26]. All item banks have shown sufficient construct validity and are now available as CAT.

Although validity of the DF-PROMIS CAT instruments has been demonstrated in clinical samples, validity alone is insufficient to justify their routine use by clinicians. For practical implementation in primary care physical therapy, test-retest reliability and minimal detectable change (MDC) are equally important. Information on the test-retest reliability and MDC of the DF-PROMIS CAT instruments is important, as low reliability or large MDCs introduce uncertainty in the scores obtained and, consequently, in the clinical decision-making for which these instruments are used [27]. Test-retest reliability refers to "the extent to which scores for patients who have not changed are the same for repeated measurement … over time" [28], while MDC represents the smallest change detectable beyond measurement error with predefined confidence [29]. Unlike classical test theory (CTT) models that produce single estimates of reliability and MDC dependent on sample characteristics, the Item Response Theory (IRT)-based DF-PROMIS CATs allow estimation of reliability and MDC at the individual patient level [30]. Clinicians thus gain the opportunity to make personalized, reliable decisions regarding patient progress and treatment effectiveness.

Therefore, the aim of the current study was to examine the test-retest reliability and MDC of the DF-PROMIS-PI v1.1, the DF-PROMIS-PF v1.2, and the DF-PROMIS-UE v2.0 in patients with musculoskeletal conditions receiving physical therapy in primary care practices.

## Methods

### Design and setting

This study was an observational cohort study. The study was performed in fourteen primary care physical therapy practices in The Netherlands. The physical therapy practices had to use the electronic health record system Fysiomanager© (Heerenveen, The Netherlands) which has an application programming interface with the Dutch-Flemish PROMIS National Center, that allows integration for administering PROMIS CATs. All participating physical therapist participated in an online training introducing the PROMIS CAT instruments and study protocol and received an instruction manual for reference. The Guidelines for Reporting Reliability and Agreement Studies (GRRAS) were used in the reporting of this work [31].

## Ethics statement

The study was reviewed by the Ethical Committee Research Healthcare Domain (ECO-GD) from the HU University of Applied Sciences Utrecht (150-000-2021) which declared this research project not subject to the Dutch Medical Research Involving Human Subjects Act and that the protocol was in accordance with the Dutch Personal Data Protection Act. The study was conducted in accordance with the Declaration of Helsinki and written informed consent was obtained from all participants by checking a box in the first digital questionnaire in a patient portal of Fysiomanager© prior to or during the first physical therapy consultation.

## Participants

Patients were recruited between July 9th, 2021, and April 30th, 2023. Patients were eligible for the study if (1) they contacted the physical therapist seeking treatment for musculoskeletal disorders of the spine or the upper extremity; and (2) were aged ≥ 18 years. Exclusion criteria were: 1) not being able to complete questionnaires; or 2) insufficient command of the Dutch language. Patients with musculoskeletal disorders of the spine or upper extremity who contacted the participating physical therapy practices were informed about the study by the physical therapist. Patients who were interested in participating were sent an information letter about the study by email and after providing informed consent, completed the first digital questionnaire in a patient portal of Fysiomanager© prior to or during the first physical therapy consultation. Patients that were not able use the online patient portal, could complete the questionnaires and the CATs during their consultations on the computer of the physical therapist. After informed consent was obtained, the physical therapist assessed eligibility for participation and informed the researchers. Due to the observational design of the study, all patients received usual physical therapy treatment.

## Sample size

The required sample size was based on the recommendations from the COSMIN-checklist for assessing the methodological quality of studies on measurement properties of PROMs [32]. The aim was to recruit patients until at least 100 patients completed the measurements at both time points for each of the DF-PROMIS CATs.

## Measures

Demographic and clinical characteristics, including age, sex, a registration code for the type of disorder, and start and end date of the physical therapy treatment series, were collected from the electronic health records. Duration of the complaint at baseline, type of disorder, educational level, and employment status were self-reported at baseline.

The PROMs used in this study were the DF-PROMIS-PI v1.1, the DF-PROMIS-PF v1.2, the DF-PROMIS-UE v2.0, and an anchor question for each of the PROMIS CAT domains. The DF-PROMIS CATs are based on IRT, and were modeled using a Graded Response Model, a generalization of the 2-parameter logistic model for dichotomous response data [18]. Each DF-PROMIS CAT begins with an item calibrated near the mean of the item bank scale, and the patient's response is used to estimate a T-score and its corresponding standard error (SE(T-score)). Subsequent most informative items are then selected iteratively based on this estimate, and the process continues until a pre-specified stopping rule is met. In this study, the stopping rules were either an SE(T-score) < 2.24 (corresponding to a reliability of approximately 0.95) with a minimum of four completed items, or a maximum of twelve items. The resulting individual scale scores are expressed as a T-score, calibrated to a mean of 50, with an SD of 10 based on the US general population [14]. This allows the patient's score to be interpreted relative to the general population, and country-specific reference values are available to assist interpretation [33]. For the items in the item banks underlying the CATs, there are three different 5-point Likert response scales: 1) unable to do/with much difficulty/with some difficulty/with a little difficulty/without any difficulty; 2) cannot do/quite a lot/somewhat/very little/not at all; and 3) cannot do because of health/a lot of difficulty/some difficulty/a little bit of difficulty/no difficulty at all. The DF-PROMIS CAT instruments do not allow missing responses.

The DF-PROMIS-PI v1.1 underlying item bank has shown good cross-cultural and construct validity [21]. The item bank contains 40 items covering a wide range of pain interference content. The time frame the patient is asked about is the past 7 days. Higher scores indicate more pain interference.

The DF-PROMIS-PF v1.2 item bank was validated in a sample of Dutch patients with chronic pain [23] and in a Dutch sample of patients receiving physical therapy [22] and showed sufficient to good psychometric properties. The item bank contains 121 items, which cover a wide range of activities, from self-care (activities of daily living) to more complex activities that require a combination of skills. The item bank includes items about functioning of the axial regions (neck and back), the upper and lower extremities, and ability to carry out instrumental activities of daily living (e.g., household chores or shopping). There is no time frame set for the items, but current status is inferred. Higher scores indicate better function.

The DF-PROMIS-UE v2.0 item bank was validated in Dutch samples of patients with upper extremity disorders, and showed sufficient psychometric properties [24–26]. The DF-PROMIS-UE v2.0 item bank contains 46 items addressing upper extremity function. Higher scores indicate better function.

The DF-PROMIS CATs were completed at baseline and a second time between three to a maximum of fourteen days after baseline measurement. This timeframe was chosen to be "long enough to prevent recall, and short enough to ensure that patients remain stable" [34], in a population of patients whose health status can change quickly for acute complaints even without treatment [35,36]. To ensure stability in the domains of interest, the patient's perceived change was assessed using an anchor question.

The anchor question related to the DF-PROMIS CAT domains were completed at the second measurement of the DF-PROMIS CATs. The anchor question for each of the DF-PROMIS CAT domains is a single item asking "To what extent do you think your pain interference/physical function/upper extremity function has changed since the start of physical therapy treatment?" with seven response options: 1) very much improved; 2) much improved; 3) somewhat improved; 4) unchanged; 5) somewhat deteriorated; 6) much deteriorated; 7) very much deteriorated.

Each patient only completed DF-PROMIS CATs and corresponding anchor questions relevant to their musculoskeletal disorder. Patients with back or neck pain completed the DF-PROMIS-PF v1.2 CAT, patients with UE disorders completed the DF-PROMIS-UE v2.0 CAT, and all patients completed the DF-PROMIS-PI v1.1 CAT.

## Data analysis

Data was analyzed using R Statistical Software (v4.3.2; R Core Team 2023) and the 'psych' package [37]. Descriptive statistics were used to explore patient characteristics. Mean and SD are reported for normally distributed data, median and inter quartile range (IQR) for non-normally distributed data.

For the assessment of test-retest reliability and the corresponding MDC, patient data was included in the analysis when the retest measurement was completed between three to fourteen days after baseline measurement, and when the patient rated "unchanged" on the domain specific anchor question.

First, CTT-based methods were used to estimate test-retest reliability and the corresponding MDC of the DF-PROMIS CATs. To this end, the intraclass correlation coefficient (ICC) was calculated using the two-way random effects model for absolute agreement [38]:

$$ICC = \frac{\sigma_p^2}{\sigma_p^2 + \sigma_m^2 + \sigma_e^2}$$

(1)

with $\sigma_p^2$ being the variation between participants, $\sigma_m^2$ being the variation between repeated measurements, and error variance $\sigma_e^2$. The two-way random effects absolute agreement ICC and the corresponding 95% confidence intervals (95%CIs) were calculated for each DF-PROMIS CAT separately. An ICC value of ≥ 0.7 is rated "sufficient" [39,40]. The MDC was also calculated for each of the DF-PROMIS CATs and was based on IRT. Since the SE(T-score) varies between

participants, the MDC was first calculated for the individual using the formula: $1.96 * \sqrt{SE(Tscore)_1^2 + SE(Tscore)_2^2}$. The MDC for the group was then obtained by calculating the mean of the individual MDCs for each of the DF-PROMIS CATs.

Although IRT models provide information on reliability in terms of the precision of the T-score (i.e., the standard error of the T-score), IRT does not provide a default method for assessing test-retest reliability. Instead, the DF-PROMIS CAT results were used to calculate participant-level reliability and MDC for the first measurement of each of the DF-PROMIS CAT instruments. In addition, the T-scores for individual participants with corresponding 95%CI at both time points were presented graphically to provide information on test-retest reliability.

For participant-level reliability, the estimate of the T-score ($\theta$) for participant p from one of the DF-PROMIS CAT instruments is represented by $\hat{\theta}_p$ with variance $\sigma_{\hat{\theta}}^2$. $SEM_p^2$ represents the squared SE(T-score), based on the IRT model. Reliability at the participant level can then be defined and calculated as [30]:

$$\rho_{\hat{\theta}_p \hat{\theta}_p} = \frac{\sigma_{\hat{\theta}}^2 - SEM_p^2}{\sigma_{\hat{\theta}}^2}$$

(2)

Because the goal is to interpret a participant's T-score relative to the reference population, the variance of the T-score of the reference population ($\sigma_\theta^2$) can be used in the calculation instead of an estimate of participant variance $\sigma_{\hat{\theta}}^2$. For the DF-PROMIS CAT instruments, the following variances from the Dutch population were used: DF-PROMIS-PF $\sigma_\theta^2 = 116.6$, DF-PROMIS-UE $\sigma_\theta^2 = 92.2$, and DF-PROMIS-PI $\sigma_\theta^2 = 74.0$ [33].

Finally, the MDC with 95% confidence at the participant level (pMDC) was calculated for the first measurement using [29]:

$$pMDC = 1.96 * \sqrt{2} * SEM_p^2$$

(3)

## Results

### Participants

A total of 1141 patients seeking treatment for musculoskeletal disorders of the spine or the upper extremity contacted the fourteen participating physical therapy practices during the study and were asked to participate. Of these patients, 67.8% (774) agreed to participate, of which 14 patients were excluded based on the exclusion criteria. Data were available for test-retest reliability analysis from 225 of the included patients: 106 completed the DF-PROMIS-PF v1.2, 93 completed the DF-PROMIS-UE v2.0, and 181 completed the DF-PROMIS-PI v1.1 (Fig 1).

The patients included in the study were mostly female (67.6%) and mean age was 47.4 (SD 15.8). Almost half of the included patients (47.3%) experienced chronic complaints. Additional demographic characteristics of the participants are shown in Table 1.

### Test-retest reliability

The group-level measures of reliability of the DF-PROMIS CATs are shown in Table 2. The DF-PROMIS CATs showed sufficient group level test-retest reliability (ICC agreement between 0.79 and 0.86). Median SE(T-score)s of the DF-PROMIS CATs ranged from 1.8 (PI domain) to 2.1 (UE domain), resulting in group-level MDC scores ranging from 4.80 to 6.08 across the respective domains..

Reliability at the participant level is plotted against the corresponding T-score for the PF-PROMIS-CAT domains in Fig 2 and shows the variability of reliability at the participant level across the observed T-scores. For observed scale scores exceeding approximately 0.5 SD above the mean (or fall below approximately 0.5 SD below the mean in the case of

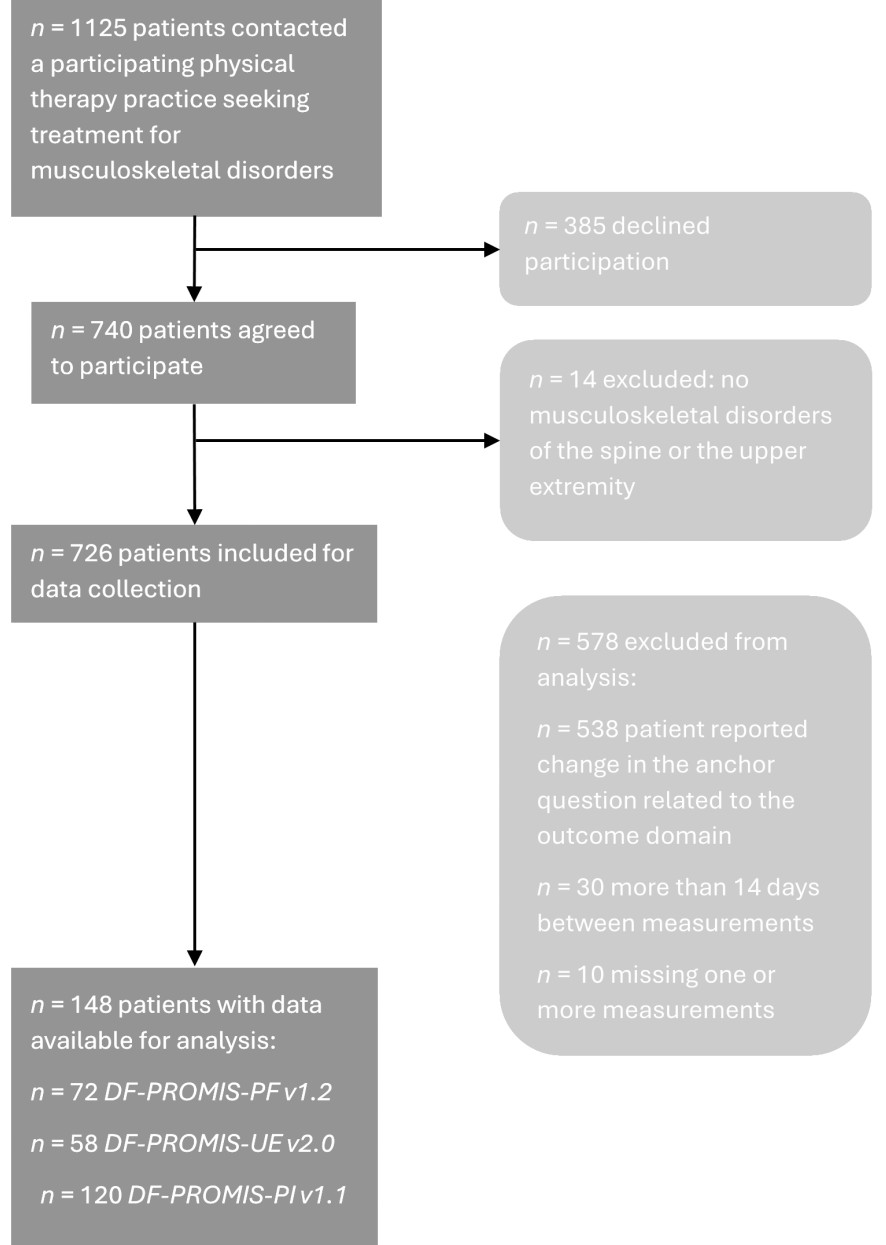

**Fig 1. Flow diagram of patient inclusion for data analysis. DF-PROMIS: Dutch-Flemish Patient Reported Outcome Measures Information System, PI: Pain Interference, PF: Physical Function, UE: Upper Extremities.**

the PI domain) reliability was below the 0.9 reliability threshold. Similarly, Fig 3 depicts the participant level MDC plotted against the observed T-scores, indicating higher MDC values for observed T-scores above the mean for the UE domain and higher MDC values for observed T-scores below the mean for the PI domain. Fig 4 displays the SE(T-score) plotted against the observed T-score, providing a clear visual representation of the impact of the stopping rules employed in the DF-PROMIS CATs. The data points on the horizontal "ceiling line" represent the stopping rule of reaching an SE(T-score) of 2.24 at or after completion of the minimum required four items, but before the twelve-item limit was reached. Data

**Table 1. Characteristics of the participants.**

| Characteristics | DF-PROMIS CAT Physical Functioning | DF-PROMIS CAT Upper Extremity | DF-PROMIS CAT Pain Interference | Overall |
|---|---|---|---|---|
| Number of patients | 106 | 93 | 181 | 225 |
| Age (years), mean (SD) | 45.6 (16.3) | 49.3 (14.5) | 47.8 (16.1) | 47.4 (15.8) |
| Sex (female), n (%) | 78 (73.6) | 55 (59.1) | 124 (68.5) | 152 (67.6) |
| Duration of current complaint, n (%) | | | | |
| Acute (<6 weeks) | 43 (40.6) | 15 (16.3) | 56 (31.1) | 68 (30.4) |
| Subacute (6–12 weeks) | 17 (16.0) | 30 (32.6) | 38 (21.1) | 50 (22.3) |
| Chronic (>12 weeks) | 46 (43.4) | 47 (51.1) | 86 (47.8) | 106 (47.3) |
| Education level, n (%) | | | | |
| Low | 17 (16.2) | 12 (13.0) | 28 (15.6) | 37 (16.6) |
| Middle | 36 (34.3) | 31 (33.7) | 56 (31.3) | 75 (33.6) |
| High | 52 (49.5) | 49 (53.3) | 95 (53.1) | 111 (49.8) |
| Employment status, n (%) | | | | |
| Full-time (>34 hours a week) | 39 (36.8) | 37 (39.8) | 66 (36.5) | 86 (38.2) |
| Part-time (12–34 hours a week) | 41 (38.7) | 34 (36.6) | 66 (36.5) | 82 (36.4) |
| Other (<12 hours a week or unpaid) | 26 (24.5) | 22 (23.7) | 49 (27.1) | 57 (25.3) |

PROMIS, Patient-Reported Outcomes Measurement Information System; CAT, Computerized Adaptive Test.

**Table 2. Test-retest results for DF-PROMIS CATs physical functioning, upper extremity, and pain interference.**

| DF-PROMIS CAT | Test mean T-score (SD) | Test median [IQR] SE(T-score) | Retest mean T-score (SD) | Retest median [IQR] SE(T-score) | ICC (95%CI) | MDC |
|---|---|---|---|---|---|---|
| DF-PROMIS CAT PF | 45.5 (6.1) | 2.1 | 45.9 (6.4) | 2.0 | 0.86 (0.8, 0.9) | 5.55 |
| DF-PROMIS CAT UE | 37.2 (8.1) | 2.1 | 37 (8.1) | 2.1 | 0.91 (0.86, 0.94) | 6.08 |
| DF-PROMIS CAT PI | 57.9 (5.5) | 1.8 | 57.5 (5.3) | 1.8 | 0.79 (0.73, 0.84) | 4.80 |

PROMIS, Patient-Reported Outcomes Measurement Information System; CAT, Computerized Adaptive Test; SD, Standard Deviation; IQR, Inter Quartile Range; ICC, Intraclass Correlation Coefficient two-way random effects model for absolute agreement; 95%CI, 95% Confidence Interval; IRT, Item Response Theory; SE(T-score), Standard Error of the T-score; MDC, Minimal Detectable Change; PF, Physical Functioning; UE, Upper Extremity; PI, Pain Interference.

points below this "ceiling line" represent participants who achieved an SE(T-score) of 2.24 or less before completing the required minimum of four items, necessitating the completion of additional items to meet the minimum required amount, which reduced the SE(T-score) further. Data points above the 2.24 "ceiling line" represent participants whose assessments were stopped because the twelve-item limit stopping rule was reached. A visual representation of test-retest reliability across the scale score range of the DF-PROMIS CAT instruments can be seen in Fig 5. The figure shows that the 95%CI of the test and retest measurements did not overlap for four, three, and eleven participants for the DF-PROMIS-PF CAT, DF-PROMIS-UE CAT, and DF-PROMIS-PI CAT respectively, which amounts to 4.7% of all observed test-retest measurements.

## Discussion

This study examined the reliability and MDC of the DF-PROMIS-PI v1.1, the DF-PROMIS-PF v1.2, and the DF-PROMIS-UE v2.0 in patients with musculoskeletal conditions receiving physical therapy in primary care practices. The results found in this study demonstrate that the test-retest reliability of all the DF-PROMIS CATs investigated exceed the threshold for "sufficient" [39,40].

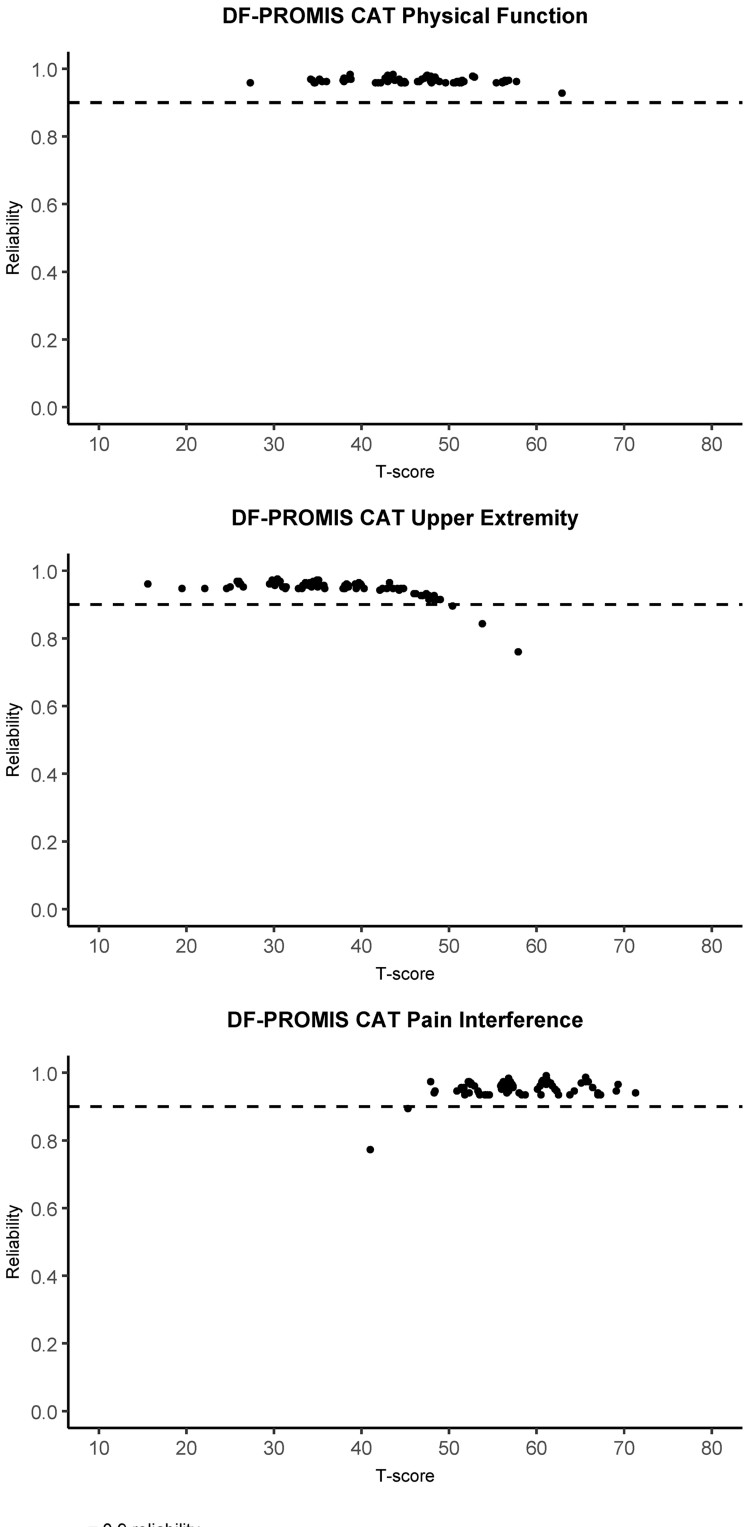

- - - - = 0.9 reliability

**Fig 2. DF-PROMIS CAT participant level reliability plotted against the t-score for the physical function, upper extremity, and pain interference domains.**

### DF-PROMIS CAT Physical Function

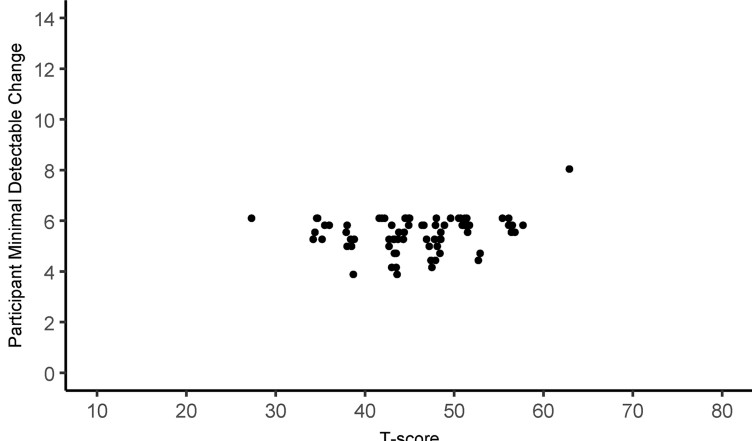

### DF-PROMIS CAT Upper Extremity

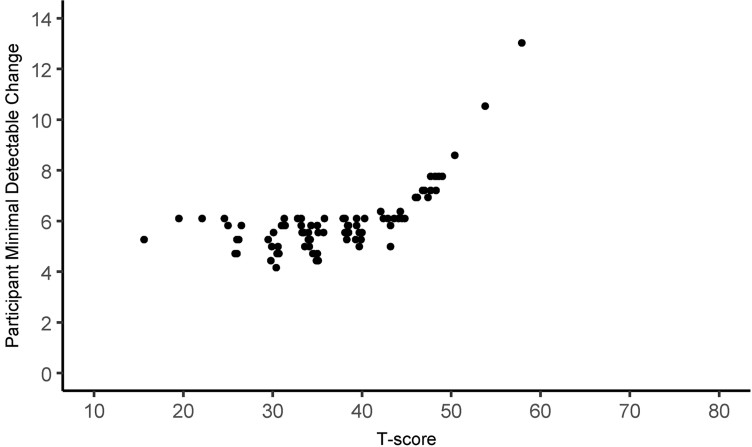

### DF-PROMIS CAT Pain Interference

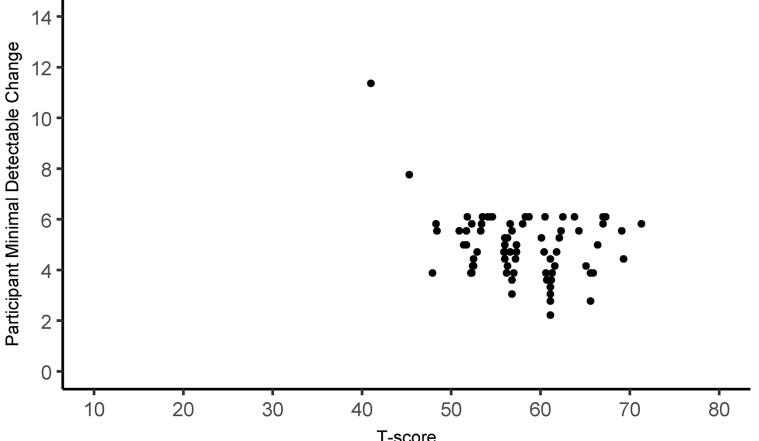

**Fig 3. DF-PROMIS CAT participant level minimal detectable change plotted against the t-score for the physical function, upper extremity, and pain interference domains.**

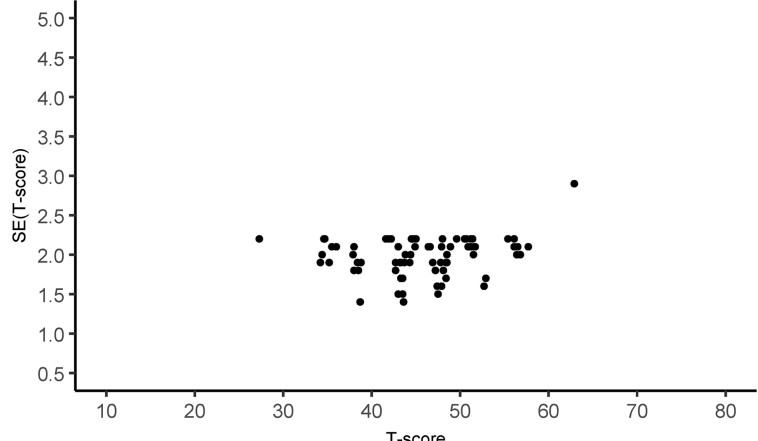

### DF-PROMIS CAT Physical Function

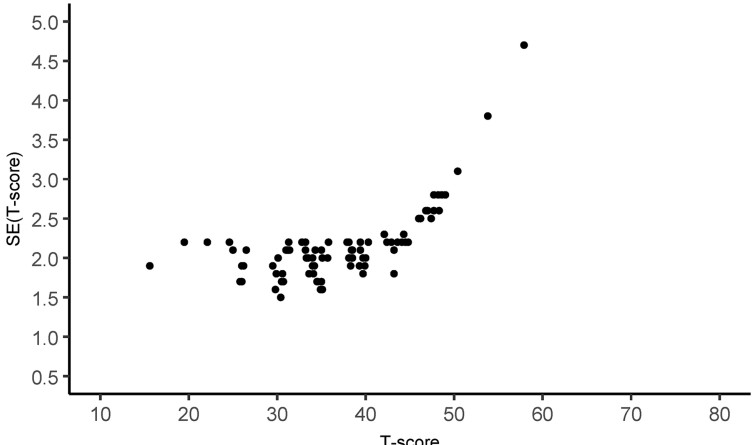

### DF-PROMIS CAT Upper Extremity

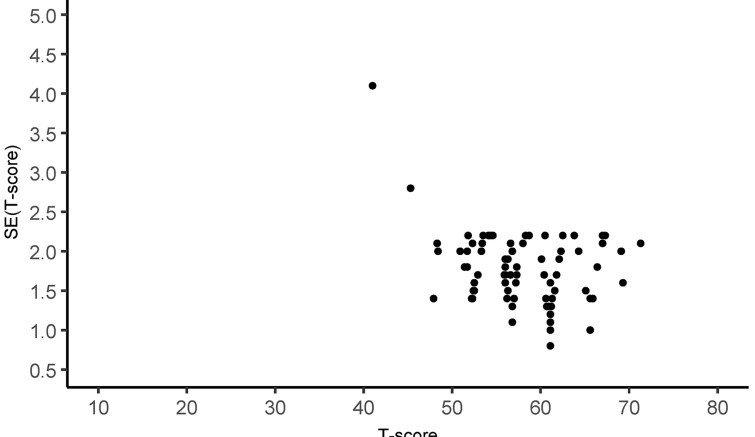

### DF-PROMIS CAT Pain Interference

**Fig 4. DF-PROMIS CAT SE(t-score) plotted against the t-score for the physical function, upper extremity, and pain interference domains. SE(t-score) of 3.16 or lower corresponds with reliability 0.9 or higher.**

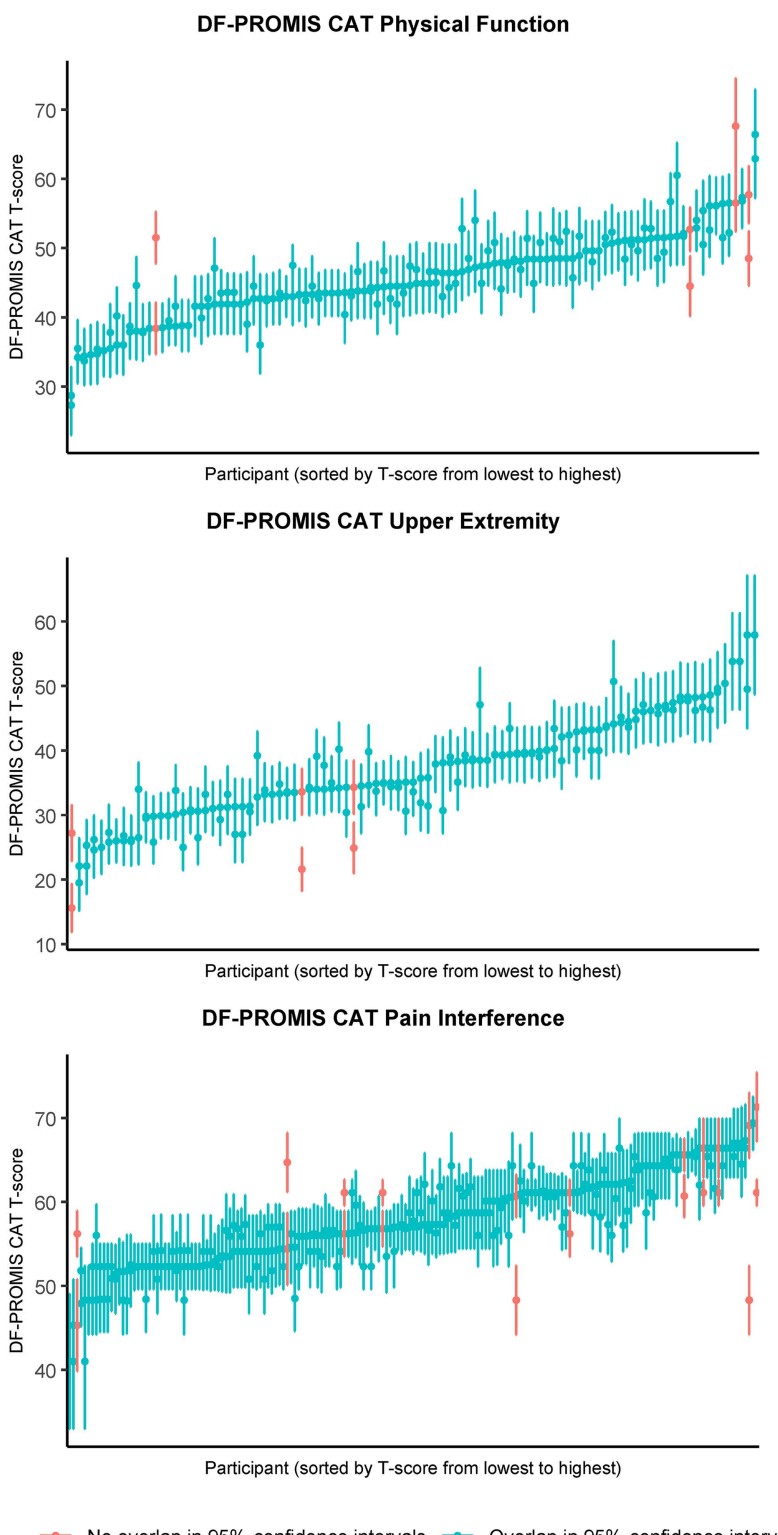

### DF-PROMIS CAT Physical Function

### DF-PROMIS CAT Upper Extremity

### DF-PROMIS CAT Pain Interference

Legend: No overlap in 95%-confidence intervals — Overlap in 95%-confidence interval

**Fig 5. A visual representation of reliability by plotting paired DF-PROMIS CAT T-scores for the test and retest measurements with the corresponding 95%CI for individual patients.**

The reliability at the participant level exceeded 0.9 for all but five of the DF-PROMIS CAT measurements, suggesting that the DF-PROMIS CATs are sufficiently precise for clinical decision-making regarding individual patients [41]. However, participant-level reliability for the UE and PI domains may fall below the 0.9 threshold as individuals' T-scores deviate further from the population mean (increasing above it for the UE domain (high level of functioning) or decreasing below it for the PI domain (low level of PI)). In practical terms, reduced precision in these instances typically does not present a problem. Likely outcomes in such scenarios include concluding that no further treatment is necessary to improve the DF-PROMIS CAT domain or that existing treatment has been effective and can be concluded. The high precision of the DF-PROMIS CATs is also reflected in the participant level MDCs (Fig 3) and group level MDCs ranging from 4.80 to 6.08.

The observed test-retest reliability outcomes at the group level align with findings from previous research investigating DF-PROMIS CATs. The DF-PROMIS-PF v1.2 CAT demonstrated sufficient test-retest reliability (ICC 0.92), mean SE(T-score) (2.06), and MDC (5.72) in Dutch patients with chronic kidney disease [42], which are comparable to those in the current study. Although no data were presented on participant-level reliability, the authors utilized the same formula to calculate the IRT-based group-level MDC. Given the mean SE(T-score) provided, it can be inferred that participant-level reliability is likely very similar to the findings of this study. Comparable results have also been reported in Dutch patients with chronic pain [23] and those receiving physical therapy [22], where similarly low SE(T-score)s (≤ 2.0) were observed.

The DF-PROMIS-UE v2.0 CAT was previously evaluated in a Dutch patients with UE disorders, where slightly higher SE(T-scores) were observed than those in the present study [26]. The patients in the current study had a higher average mean T-score of 37.2 compared to the earlier cohort (range 33.4–34.7), which might account for this difference. However, the difference is relatively minor and is unlikely to influence clinical decisions at the individual level. Wilkinson et al. (2021) reported an ICC of 0.82 with an 83% confidence interval (CI) ranging from 0.77 to 0.86, slightly lower than the ICC of 0.91 with a 95% CI ranging from 0.86 to 0.94 found in the current study [43]. Despite the different thresholds used for calculating the CIs, their similarity further supports consistency in the ICCs reported.

Participant-level reliability of the DF-PROMIS-PI v1.1 CAT has shown to be > 0.95 in Dutch patients with chronic pain [21], Dutch patients with Rheumatoid Arthritis [44], and Dutch patients with chronic kidney disease [42] with T-scores ranging from 50 to 80. These observations are almost identical to the findings in the current study. Mean T-scores are also very similar between the different patient groups investigated, with the exception of the Dutch patients with chronic pain with a mean T-score of 64.1 [21].

Collectively, the DF-PROMIS PF, UE, and PI domain CATs consistently demonstrate sufficient reliability and precision for measuring their respective domains in both clinical practice and research settings across different patient groups. [45].. The consistency of precision across different populations underscores a significant advantage of IRT-based PROMs over CTT-based PROMS, namely that the precision of T-scores is an inherent characteristic of the instrument, independent of the sample.

The DF-PROMIS PF, UE, and PI domain CATs also show consistent MDCs at both the group and participant level across patient populations, as shown by the similar SE(T-scores). This similarity is unsurprising, since the SE(T-score) depends on the location of the T-score across the scale of the CAT, with higher SE(T-scores) at the extremes and the lowest SE(T-scores) around the mean of the scale [46]. Therefore, patient groups with similar average ability (T-score) in a DF-PROMIS domain have similar mean SE(T-score)s and consequently similar MDC values.

The test-retest reliability of the DF-PROMIS CATs examined in this study, assessed using CTT-based methods, also demonstrates sufficient reliability across all three domains, with ICC scores ranging from 0.79 to 0.91. These findings align with reliability results from domain-specific PROMs. For example, the Quebec Back Pain Disability Scale [47] (QBPDS) as a measure of physical function in patients with lower back pain (LBP) was found to have sufficient reliability, ranging from 0.70 to 0.99 [48]. Similarly, the Neck Disability Index [49] (NDI), a measure of physical function in patients with neck pain, showed a pooled ICC of 0.91 [50]. Although the latter is higher, it is similar to the ICC of 0.86 found in this study for physical function. Test-retest reliability based on CTT methods for the DF-PROMIS UE domain CAT can be compared with the

findings from the Disabilities of the Arm, Shoulder, and Hand [51] (DASH) questionnaire. Reliability of the DASH has been shown to range from 0.91 to 0.96 in patients with shoulder disorders [52]. These findings are again very similar to the ICC of 0.91 for the DF-PROMIS-UE v2.0 CAT found in the current study.

For clinical care, the observed reliability and low MDCs indicate that clinicians can trust PROMIS CAT scores to be precise and reproducible across diverse musculoskeletal populations. This supports their use for monitoring outcomes and guiding treatment decisions at both the individual and group level. Because DF-PROMIS CATs adaptively administer only the most informative items, they reduce patient burden while maintaining measurement precision, making them feasible for integration into routine primary care physical therapy. Furthermore, the standardized T-score metric facilitates comparisons across conditions and patient groups, allowing clinicians to interpret outcomes consistently in heterogeneous patient populations. Together, these properties strengthen the case for wider implementation of PROMIS CATs in physical therapy where efficient, low-burden, and clinically meaningful outcome measurement is needed [53].

## Strengths and limitations

Several strengths and limitations must be considered when interpreting the results of this study. A notable strength is the large sample sizes for all three DF-PROMIS domains [32]. Meeting these recommendations enhances confidence in the results and strengthens the methodological quality of the study. Although the threshold of 100 patients was not reached for the UE domain (93 patients), the impact on the generalizability of the results is likely minor. Additionally, the study's design minimized the effort required from physical therapists for patient recruitment and data collection, and combined with broad eligibility criteria, ensured that the included patients closely reflect the average patient treated by physical therapists. Furthermore, the inclusion of an anchor question asking patients to report changes on the DF-PROMIS domains ensured that only data from patients without changes were included in the analysis.

A limitation to consider is the large number of patients excluded from the analysis due to reporting change on the anchor questions, exceeding the maximum number of days between the two measurements, or missing data for the baseline or follow-up measurement (535 patients). This exclusion necessitated the recruitment of additional patients to meet the sample size requirements, leading to a waste of time and resources, and might have introduced bias. Another potential source of bias is the significant number of patients who declined to participate. To keep data collection feasible and minimize the burden on participants, reasons for non-participation were not recorded, leaving it unknown if and how much bias was introduced. The final limitation is that CATs require an electronic device, such as a smartphone, tablet, or personal computer, for completion. This requirement might exclude patients lacking digital skills from completing the DF-PROMIS CATs. To mitigate this issue, patients could complete the measurements during a treatment session with their physical therapist, although this could potentially lead to biased responses. Unfortunately, data collection did not include whether the patient completed the measurements alone or with support from their physical therapist.

A final important consideration is that the DF-PROMIS-PI uses a 7-day recall period, whereas the DF-PROMIS-PF and DF-PROMIS-UE capture current function. For the DF-PROMIS-PI, a 3-day retest interval may lead to partial overlap of recall periods, potentially inflating agreement. In contrast, DF-PROMIS-PF and DF-PROMIS-UE scores reflect the patient's status at the time of measurement, making their reliability more directly dependent on short-term stability. Clinically, this implies that when follow-up measurements with the DF-PROMIS-PI are planned within seven days, the recall period would need to be adjusted to prevent overlap; however, any modification of the recall period requires prior approval from HealthMeasures or the DF-PROMIS National Center.

The DF-PROMIS CATs require fewer items to complete, are more accurate than legacy instruments, allow comparison between different patient groups relative to the reference population on a domain, and adapt to each individual patient based on their responses to the items in the DF-PROMIS CAT. Future research should focus on implementing the DF-PROMIS CATs in clinical practice by investigating their responsiveness and clinical feasibility. Researchers should

consider adopting the DF-PROMIS domains as outcomes for intervention studies to facilitate interpretability and comparison of results between different patient groups.

## Conclusion

The DF-PROMIS PF, UE, and PI CATs demonstrated sufficient reliability and precision at both the group-level and participant-level in patients with musculoskeletal conditions receiving physical therapy in primary care practices. Future research should focus on implementing DF-PROMIS CATs in clinical practice, examining their responsiveness, and evaluate their feasibility. Emphasizing the adoption of DF-PROMIS domains as outcomes in intervention studies and clinical practice will enhance the interpretability and comparability of results across different patient groups, thereby potentially improving patient care and outcomes in physical therapy.

## Author contributions

**Conceptualization:** E.J.A. Haan, C.B. Terwee, H. Wittink, H. Kiers.

**Data curation:** R.M. Arensman, J. van Rosmalen.

**Formal analysis:** R.M. Arensman, J. van Rosmalen.

**Funding acquisition:** C.B. Terwee, H. Wittink, H. Kiers.

**Investigation:** E.J.A. Haan.

**Methodology:** E.J.A. Haan, C.B. Terwee, J. van Rosmalen, H. Wittink, H. Kiers.

**Project administration:** E.J.A. Haan, H. Wittink, H. Kiers.

**Resources:** C.B. Terwee, H. Wittink, H. Kiers.

**Supervision:** C.B. Terwee, H. Wittink, H. Kiers.

**Validation:** R.M. Arensman.

**Visualization:** R.M. Arensman.

**Writing – original draft:** R.M. Arensman.

**Writing – review & editing:** R.M. Arensman, E.J.A. Haan, C.B. Terwee, J. van Rosmalen, H. Wittink, H. Kiers.

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
