## [Decision Letter · Decision Letter 0]

15 Jul 2025

Dear Dr. Arensman,

Thank you for submitting your manuscript to PLOS ONE. After careful consideration, we feel that it has merit but does not fully meet PLOS ONE’s publication criteria as it currently stands. Therefore, we invite you to submit a revised version of the manuscript that addresses the points raised during the review process.

We look forward to receiving your revised manuscript.

Kind regards,

Mark Hwang

Academic Editor

PLOS ONE

Journal Requirements:

2. Thank you for stating the following financial disclosure: [This study is co-funded by the Taskforce for Applied Research SIA (RAAK.MKB13.025), part of the Dutch Research Council (NWO).]

3. Thank you for stating the following in the Competing Interests section: [I have read the journal's policy and the authors of this manuscript have the following competing interests: author C.B. Terwee, PhD, is past board member of the PROMIS Health Organization and representative of the Dutch-Flemish PROMIS National Center.].

5. Please include captions for your Supporting Information files at the end of your manuscript, and update any in-text citations to match accordingly. Please see our Supporting Information guidelines for more information: http://journals.plos.org/plosone/s/supporting-information .

Reviewers' comments:

Reviewer's Responses to Questions

**Comments to the Author**

1. Is the manuscript technically sound, and do the data support the conclusions?

Reviewer #1: Yes

Reviewer #2: Yes

2. Has the statistical analysis been performed appropriately and rigorously?

Reviewer #1: I Don't Know

Reviewer #2: Yes

3. Have the authors made all data underlying the findings in their manuscript fully available?

Reviewer #1: Yes

Reviewer #2: Yes

4. Is the manuscript presented in an intelligible fashion and written in standard English?

Reviewer #1: No

Reviewer #2: Yes

Reviewer #1: This manuscript addresses the test-retest reliability and minimal detectable change (MDC) of the Dutch-Flemish PROMIS CATs in patients with musculoskeletal conditions in primary care physical therapy. The topic is timely and clinically relevant, and the use of both CTT and IRT-based approaches adds methodological depth. However, several issues require attention.

1. The introduction could be more concise and improved in flow, especially in the first two paragraphs.

A. The transition between the general use of PROMs and the introduction of PROMIS (lines 76–77) could be smoother. Explicitly stating why PROMIS was selected over other PROMs would strengthen the rationale.

B. Lines 63–66 introduce the biopsychosocial model and patient-centered care. This is important, but the link to PROMs and clinical reasoning could be better framed by briefly explaining how PROMs operationalize this model.

C. The description of CAT functioning (lines 84–90) is quite technical for an introduction. While this detail is useful, consider whether some of it might be moved to the Methods section or summarized more succinctly here.

D. The final paragraph (lines 104–116) clearly states the study aim. You might enhance the impact by briefly reinforcing why test-retest reliability and MDC are crucial for clinical uptake—especially in the Dutch primary care setting where time and efficiency are key concerns.

2. Justification of test-retest interval. The rationale for the short lower bound of 3 days is missing. This short interval raises concerns about recall bias. The authors should either justify this choice with references or revise the interpretation of reliability results accordingly. This should also be adressed in the discussion.

3. The PROMIS-PI uses a 7-day recall period, while the PROMIS-PF does not specify a time frame. It may be worth briefly reflecting on how this discrepancy could influence the interpretation of test-retest reliability.

4. While the discussion provides a solid overview of the findings and compares them to previous research, it currently lacks depth in interpreting what the results actually mean for clinical practice. At present, it reads primarily as a summary of results and literature alignment. Rather than simply stating that outcomes are “comparable,” I recommend the authors elaborate on the significance of the results. For example in the paragraph starting at line 304: Does this alignment suggest the DF-PROMIS CATs perform consistently across diverse patient populations? How might these psychometric properties support the generalizability or robustness of the tool in real-world clinical settings? Could this strengthen the argument for wider implementation of PROMIS CATs in physical therapy, especially when considering resource constraints or variability in clinician expertise? Adding this kind of interpretative commentary would help move the current discussion beyond data repetition and make it more meaningful for readers and clinicians. Please do this throughtout the whole discussion, not only paragraph 3.

Reviewer #2: Overall, looking good. Prior publications have shown that PROMIS may be used to assess symptoms in rheumatological diseases, for example Ankylosing spondylitis and Rheumatoid arthritis. It is good to know that the Dutch-Flemish translation is working well.

**Do you want your identity to be public for this peer review?** For information about this choice, including consent withdrawal, please see our Privacy Policy

Reviewer #1: No

Reviewer #2: No

---

## [Author Response · Author response to Decision Letter 1]

27 Aug 2025

Dear editor and reviewers,

We would like to thank you for your constructive feedback on our manuscript. Your comments have helped us to clarify our arguments, refine the discussion of our findings, and strengthen the overall quality of the paper. We have carefully considered each suggestion and revised the manuscript accordingly, which we believe has led to substantial improvements.

Below you will find our point by point response and the changes we made to the manuscript.

Journal Requirements:

Response by the authors: We made several changes to the style, layout, and references following a review of the guidelines.

2. Thank you for stating the following financial disclosure: [This study is co-funded by the Taskforce for Applied Research SIA (RAAK.MKB13.025), part of the Dutch Research Council (NWO).]

Response by the authors: We added the role of the funder statement to the cover letter.

3. Thank you for stating the following in the Competing Interests section: [I have read the journal's policy and the authors of this manuscript have the following competing interests: author C.B. Terwee, PhD, is past board member of the PROMIS Health Organization and representative of the Dutch-Flemish PROMIS National Center.].

Response by the authors: We included the updated Competing Interests statement in the cover letter.

Response by the authors: We have included a full ethics statement in the Methods section of the manuscript.

Response by the authors: Our manuscript does not include supporting information.

Response by the authors: Not applicable since the reviewers did not recommend the inclusion of specific previously published works.

Reviewers' comments:

Reviewer's Responses to Questions

Comments to the Author

1. Is the manuscript technically sound, and do the data support the conclusions?

Reviewer #1: Yes

Reviewer #2: Yes

2. Has the statistical analysis been performed appropriately and rigorously?

Reviewer #1: I Don't Know

Reviewer #2: Yes

3. Have the authors made all data underlying the findings in their manuscript fully available?

The PLOS Data policy requires authors to make all data underlying the findings described in their manuscript fully available without restriction, with rare exception (please refer to the Data Availability Statement in the manuscript PDF file). The data should be provided as part of the manuscript or its supporting information, or deposited to a public repository. For example, in addition to summary statistics, the data points behind means, medians and variance measures should be available. If there are restrictions on publicly sharing data—e.g. participant privacy or use of data from a third party—those must be specified.Reviewer #1: Yes

Reviewer #2: Yes

4. Is the manuscript presented in an intelligible fashion and written in standard English?

Reviewer #1: No

Reviewer #2: Yes

5. Review Comments to the Author

Reviewer #1: This manuscript addresses the test-retest reliability and minimal detectable change (MDC) of the Dutch-Flemish PROMIS CATs in patients with musculoskeletal conditions in primary care physical therapy. The topic is timely and clinically relevant, and the use of both CTT and IRT-based approaches adds methodological depth. However, several issues require attention.

1. The introduction could be more concise and improved in flow, especially in the first two paragraphs.

A. The transition between the general use of PROMs and the introduction of PROMIS (lines 76–77) could be smoother. Explicitly stating why PROMIS was selected over other PROMs would strengthen the rationale.

B. Lines 63–66 introduce the biopsychosocial model and patient-centered care. This is important, but the link to PROMs and clinical reasoning could be better framed by briefly explaining how PROMs operationalize this model.

C. The description of CAT functioning (lines 84–90) is quite technical for an introduction. While this detail is useful, consider whether some of it might be moved to the Methods section or summarized more succinctly here.

D. The final paragraph (lines 104–116) clearly states the study aim. You might enhance the impact by briefly reinforcing why test-retest reliability and MDC are crucial for clinical uptake—especially in the Dutch primary care setting where time and efficiency are key concerns.

Response by the authors: To improve the flow of the introduction and to make it more concise, we added justification for the choice for PROMIS, briefly expanded on the link between PROMs and clinical reasoning/clinical practice, and we moved part of the technical description of the functioning of the CATs to the methods. We also added a sentence reinforcing the importance of reliability and MDC for clinical use.

Changes made:

Added and rewrote lines 66 to 72: Patient Reported Outcome Measures (PROMs) help to operationalize these principles by systematically capturing the patient’s own perspective on their symptoms, functioning, and quality of life, thereby ensuring that biological, psychological, and social dimensions of health are included in clinical decision-making [8]. In this way, PROMs support physical therapists in their clinical reasoning related to diagnosis, treatment, and evaluation To assist physical therapists in their clinical reasoning related to diagnosis, treatment, and evaluation, Patient Reported Outcomes Measures (PROMs) can provide valuable insight from the patient’s perspective [8]. A recent systematic review found evidence that feedback from PROMs can improve quality of life, patient-provider communication, and disease control PROMs capture the patient’s perspective on their condition, treatment, and its impact on their life. A recent systematic review found evidence that feedback from PROMs can improve quality of life, patient-provider communication, and disease control [9].

Added and rewrote lines 81-83: PROMIS instruments were selected because they provide precise and efficient assessment of relevant health outcomes, with standardized scores that enable comparability across studies and patient populations [14,17].

Rewrote lines 86-93 to: From these item banks, Computerized Adaptive Tests (CATs) [19] were developed, which select the most informative questions for each individual based on their responses. This results in patient-specific assessments that require fewer items to complete, while achieving equal or greater measurement precision compared to traditional fixed item questionnaires[14,20]. PROMIS scores are reported as T-scores, standardized to a reference population with a mean of 50 and a standard deviation of 10 [18]. This allows patients’ outcomes to be interpreted relative to the general population and facilitates comparisons across different patient groups using a common metric.

Added lines 104-106: Information on the test-retest reliability and MDC of the DF-PROMIS CAT instruments is important, as low reliability or large MDCs introduce uncertainty in the scores obtained and, consequently, in the clinical decision-making for which these instruments are used [27].

Rewrote lines 159-171: The PROMs used in this study were the DF-PROMIS-PI v1.1, the DF-PROMIS-PF v1.2, the DF-PROMIS-UE v2.0, and an anchor question for each of the PROMIS CAT domains. The DF-PROMIS CATs are based on IRT, and were modeled using a Graded Response Model, a generalization of the 2-parameter logistic model for dichotomous response data [18]. Each DF-PROMIS CAT begins with an item calibrated near the mean of the item bank scale, and the patient’s response is used to estimate a T-score and its corresponding standard error (SE(T-score)). Subsequent most informative items are then selected iteratively based on this estimate, and the process continues until a pre-specified stopping rule is met. In this study, the stopping rules were either an SE(T-score) < 2.24 (corresponding to a reliability of approximately 0.95) with a minimum of four completed items, or a maximum of twelve items. The resulting individual scale scores are expressed as a T-score, calibrated to a mean of 50, with an SD of 10 based on the US general population [14]. This allows the patient’s score to be interpreted relative to the general population, and country-specific reference values are available to assist interpretation [33].

2. Justification of test-retest interval. The rationale for the short lower bound of 3 days is missing. This short interval raises concerns about recall bias. The authors should either justify this choice with references or revise the interpretation of reliability results accordingly. This should also be adressed in the discussion.

Response by the authors: The retest interval was set at a lower bound of 3 days to balance the competing risks of recall bias and true change in the domains we measured. The COSMIN Study Design checklist for Patient-reported outcome measurement instruments (Mokkink et al. 2019) recommends to “use an appropriate time interval between the two measurements, which is long enough to prevent recall, and short enough to ensure that patients remain stable” for the assessment of test-retest reliability. In our population of interest (patients with MSK), symptoms such as pain and function can fluctuate rapidly, making longer retest intervals (e.g., ≥7 days) more likely to capture true clinical change rather than measurement error. Conversely, very short intervals (e.g., <24 hours) may increase the risk of recall effects. We therefore considered a 3-day period optimal to minimize the likelihood of substantial health status change while reducing the potential for patients to remember their initial responses. In addition, patients were asked at retest whether their complaints had changed and those reporting change were excluded from reliability analyses to ensure stability in the domain of interest. We added a short clarification for the choice of the timeframe in the methods.

Changes made: Added to lines 191 – 195: This timeframe was chosen to be “long enough to prevent recall, and short enough to ensure that patients remain stable” [34], in a population of patients whose health status can change quickly for acute complaints even without treatment [35,36]. To ensure stability in the domains of interest, the patient’s perceived change was assessed using an anchor question.

3. The PROMIS-PI uses a 7-day recall period, while the PROMIS-PF does not specify a time frame. It may be worth briefly reflecting on how this discrepancy could influence the interpretation of test-retest reliability.

Response by the authors: We agree that this difference in recall period between the DF-PROMIS instruments warrants a brief reflection and we added this in the discussion.

Changes made: Added a paragraph to the discussion lines 395-405: A final important consideration is that the DF-PROMIS-PI uses a 7-day recall period, whereas the DF-PROMIS-PF and DF-PROMIS-UE capture current function. For the DF-PROMIS-PI, a 3-day retest interval may lead to partial overlap of recall periods, potentially inflating agreement. In contrast, DF-PROMIS-PF and DF-PROMIS-UE scores reflect the patient’s status at the time of measurement, making their reliability more directly dependent on short-term stability. Clinically, this implies that when follow-up measurements with the DF-PROMIS-PI are planned within seven days, the recall period would need to be adjusted to prevent overlap; however, any modification of the recall period requires prior approval from HealthMeasures or the DF-PROMIS National Center.

4. While the discussion provides a solid overview of the findings and compares them to previous research, it currently lacks depth in interpreting what the results actually mean for clinical practice. At present, it reads primarily as a summary of results and literature alignment. Rather than simply stating that outcomes are “comparable,” I recommend the authors elaborate on the significance of the results. For example in the paragraph starting at line 304: Does this alignment suggest the DF-PROMIS CATs perform consistently across diverse patient populations? How might these psychometric properties support the generalizability or robustness of the tool in real-world clinical settings? Could this strengthen the argument for wider implementation of PROMIS CATs in physical therapy, especially when considering resource constraints or variability in clinician expertise? Adding this kind of interpretative commentary would help move the current discussion beyond data repetition and make it more meaningful for readers and clinicians. Please do this throughtout the whole discussion, not only paragraph 3.

Response by the authors: We understand the feedback from the reviewer. Instead of adding this information throughout the discussion, we added a dedicated paragraph to maintain readab

---

## [Decision Letter · Decision Letter 1]

17 Sep 2025

Test-retest reliability and minimal detectable change of Dutch-Flemish Patient Reported Outcomes Measurement Information System (PROMIS®) Computerized Adaptive Tests for musculoskeletal disorders.

PONE-D-25-27506R1

Dear Dr.Arensman

We’re pleased to inform you that your manuscript has been judged scientifically suitable for publication and will be formally accepted for publication once it meets all outstanding technical requirements.

Kind regards,

Mark Hwang

Academic Editor

PLOS ONE

Additional Editor Comments (optional):

Reviewer #1:

Reviewer #2:

Reviewers' comments:

Reviewer's Responses to Questions

**Comments to the Author**

Reviewer #1: All comments have been addressed

Reviewer #2: All comments have been addressed

2. Is the manuscript technically sound, and do the data support the conclusions?

Reviewer #1: Yes

Reviewer #2: Yes

3. Has the statistical analysis been performed appropriately and rigorously?

Reviewer #1: Yes

Reviewer #2: Yes

4. Have the authors made all data underlying the findings in their manuscript fully available?

Reviewer #1: (No Response)

Reviewer #2: Yes

5. Is the manuscript presented in an intelligible fashion and written in standard English?

Reviewer #1: Yes

Reviewer #2: Yes

Reviewer #1: I would like to thank the authors for addressing the comments and revising the manuscript accordingly. At this time I have no further remarks. I recommend acceptance of the manuscript in its current form.

Reviewer #2: Overall, looking good. Prior publications have shown that PROMIS may be

used to assess symptoms in rheumatological diseases, for example Ankylosing spondylitis

and Rheumatoid arthritis. It is good to know that the Dutch-Flemish translation is working

well.

**Do you want your identity to be public for this peer review?** For information about this choice, including consent withdrawal, please see our Privacy Policy

Reviewer #1: No

Reviewer #2: No

---

## [Editor Report · Acceptance letter]

PONE-D-25-27506R1

PLOS ONE

Dear Dr. Arensman,

I'm pleased to inform you that your manuscript has been deemed suitable for publication in PLOS ONE. Congratulations! Your manuscript is now being handed over to our production team.

Kind regards,

on behalf of

Dr. Mark Hwang

Academic Editor

PLOS ONE